# Fascial Innervation: A Systematic Review of the Literature

**DOI:** 10.3390/ijms23105674

**Published:** 2022-05-18

**Authors:** Vidina Suarez-Rodriguez, Caterina Fede, Carmelo Pirri, Lucia Petrelli, Juan Francisco Loro-Ferrer, David Rodriguez-Ruiz, Raffaele De Caro, Carla Stecco

**Affiliations:** 1Doctorate School, University of Las Palmas de Gran Canaria, 35001 Las Palmas de Gran Canaria, Spain; 2Department of Neurosciences, Institute of Human Anatomy, University of Padova, 35121 Padova, Italy; caterina.fede@unipd.it (C.F.); carmelop87@hotmail.it (C.P.); lucia.petrelli@unipd.it (L.P.); rdecaro@unipd.it (R.D.C.); carla.stecco@unipd.it (C.S.); 3Department of Clinical Sciences, University of Las Palmas de Gran Canaria, 35016 Las Palmas de Gran Canaria, Spain; juanfrancisco.loro@ulpgc.es; 4Department of Physical Education, University of Las Palmas de Gran Canaria, 35017 Las Palmas de Gran Canaria, Spain; david.rodriguezruiz@ulpgc.es

**Keywords:** fascia, innervation, nociceptor, pain

## Abstract

Currently, myofascial pain has become one of the main problems in healthcare systems. Research into its causes and the structures related to it may help to improve its management. Until some years ago, all the studies were focused on muscle alterations, as trigger points, but recently, fasciae are starting to be considered a new, possible source of pain. This systematic review has been conducted for the purpose of analyze the current evidence of the muscular/deep fasciae innervation from a histological and/or immunohistochemical point of view. A literature search published between 2000 and 2021 was made in PubMed and Google Scholar. Search terms included a combination of fascia, innervation, immunohistochemical, and different immunohistochemical markers. Of the 23 total studies included in the review, five studies were performed in rats, four in mice, two in horses, ten in humans, and two in both humans and rats. There were a great variety of immunohistochemical markers used to detect the innervation of the fasciae; the most used were Protein Gene Marker 9.5 (used in twelve studies), Calcitonin Gene-Related Peptide (ten studies), S100 (ten studies), substance P (seven studies), and tyrosine hydroxylase (six studies). Various areas have been studied, with the thoracolumbar fascia being the most observed. Besides, the papers highlighted diversity in the density and type of innervation in the various fasciae, going from free nerve endings to Pacini and Ruffini corpuscles. Finally, it has been observed that the innervation is increased in the pathological fasciae. From this review, it is evident that fasciae are well innerved, their innervation have a particular distribution and precise localization and is composed especially by proprioceptors and nociceptors, the latter being more numerous in pathological situations. This could contribute to a better comprehension and management of pain.

## 1. Introduction

These days, myofascial pain syndromes represent a very important issue in the healthcare, since the number of people who are suffering from them is increasing, and this situation has an enormous impact not only on individuals, but on society. In fact, the diagnosis is often dismissed by clinicians, while patients remain in pain for years, and effective treatments are lacking due to the absence of a clear mechanism. Therefore, it is necessary to deepen the knowledge with the aim of developing appropriate prevention and management strategies. A key point is the possible role of fasciae in the perception of pain. In the past, fascia has been generally considered as an inert wrapping organ that gave mechanical support to our muscles and other organs. Existing histological research recognizes the presence of sensory nerves in fascia [1,2], but they were not given due consideration, especially in relation to the understanding of musculoskeletal dynamics. Whereas the developers of related somatic therapies, Moshe Feldenkrais and Ida Rolf, were seemingly unaware of the importance of the fascia as a sensory organ, the osteopathy creator, Andrew Taylor Still, stated that, ‘No doubt nerves exist in the fascia …’ and recommended that all fascial tissues should be treated with the same degree of respect as if dealing with ‘the branch offices of the brain’ [3]. Observations of a rich presence of sensory nerves in fascial tissues were independently announced by three teams from different countries in the first international Fascial Research Congress [4]. Since then, several investigations have been performed and published about fascial innervation, proposing that the fasciae may be considered our largest sensory organ given its complete surface area, as well as participating actively in proprioception and nociception. In recent years, various research studies were published about this topic, and it seems that different fasciae have different type and density of innervation, showing that these tissues are more complex than someone would think. Hence, the purpose of this review is to collect, systematically present, and analyze the existing evidence on fascial innervation and its possible correlation with pain. For this review, we decided to focus only on the muscular/deep fascia and to consider only the histological and immunohistochemical studies, and not in the clinical studies, to understand how strong our knowledge about fascial innervation and what fasciae of the body have been studied to date. In addition, we included studies related to the retinacula, as they are reinforcements of the muscular fascia, and periosteum, as it is in continuity with the muscular fascia, and shares with it the same anatomical features [5].

Studying fascial innervation could provide new insight into pain generation to support clinicians to better understand the origin of certain types of pain and to focus on more specific treatments.

## 2. Materials and Methods

The electronic databases PUBMED and GOOGLE SCHOLAR were used to develop the search, the term “fascia” was used in combination with the standard operator “AND” with the following terms: “innervation”, “immunochemistry”, “immunohistochemistry”, “immunocytochemistry” “nerve fibers”, “PGP”, “TH”, “S100”, “SP”, “CGRP”, “NGF”, “neurofilament protein”, “DiI”, “NF200”, “MBP”, “peripherin”, “NSE”, “TRPV1”, “RT97”, “tubulin”, “NR2B”. Limitations included English and Spanish language and years 2000–2021. First, they were selected by title and then, by reading the abstracts. The inclusion criteria were studies with animals and humans, studies that included any nerve receptor in the fascia, original papers, and studies including the innervation of the retinacula, considering them as fascial reinforcements. The exclusion criteria were studies not related to the fascia, clinical studies, studies on ligaments, tendons and/or visceral fasciae. The initial search identified 5210 articles with 410 manually selected by title, 40 by abstract, 22 eliminated based on exclusion criteria (Figure 1). Since the refined search did show few results, references from revisions found in the search were hand-searched to identify other potentially eligible studies for inclusion in the analysis, missed by the initial search. In such a way, other 8 papers were added to the review. Twenty-three articles underwent full-text reviews. The data that were extracted, independently by two of the researchers, from the reviewed articles were: type and size of the sample, type of immunohistochemical markers, type and location of nerve receptors.

## 3. Results

### 3.1. Sample

Of the 23 total studies, five studies were performed in rats [7,8,9,10,11], four in mice [7,8,9,10], two in horses [11,12], ten in humans, among which there were six in human cadavers [13,14,15,16,17,18], two in live humans [19], and two in human cadavers and live humans [20,21]. Two studies were made both in humans and rats [22,23].

### 3.2. Type of Immunohistochemical Markers

A variety of antibodies were used for immunodetection: Protein Gene Marker 9.5 (PGP 9.5) was used in twelve studies; Calcitonin Gene-Related Peptide (CGRP) in ten: S100 in ten; Substance P (SP) in seven; Tyrosine Hydroxylase (TH) in six; Nerve Growth Factor (NGF) in three; Neurofilament (NF200) in two; Transient Receptor Potential Cation Channel Subfamily V Member 1 (TRPV1) in two; Neurofilament protein, DiI, Myelin Basic Protein (MBP), Peripherin, Neuron-Specific Enolase (NSE), Neurofilament Monoclonal Antibody (RT97), Tubulin, and NR2B each in one. Table 1 summarizes the kind of markers and what they were used for in every study.

### 3.3. Type and Localization of Nerve Receptors

In order to carry out effective work, the information retrieved was classified according to the following body parts: head, trunk, superior limb and inferior limb. Then, the innervation found in the specifics areas of these parts was compiled and summarized.

In the head, there is only one study on the masseter fascia [26]. These authors compared the innervation density between the muscle and its fascia, demonstrating that the latter is more innerved (404.5 fibers/area mm^2^ inside the connective tissue) with respect to the muscle (227.6 fibers/area mm^2^). Besides, most of the nerves were nociceptors, 57.6% fibers expressing SP alone, 29.1% SP/NR2B, 9.2% SP/NGF and 8% SP/NR2B/NGF in combination.

Related to the trunk, the fasciae studied were the trapezius, pectoralis, gluteal fasciae, and the thoracolumbar fascia (TLF). There are eight studies about the TLF, resulting the most studied fascia. All of them found free nerve endings running through various layers of the TLF. However, the density of innervation among them seems different, specifically, CGRP immunoreactive (IR) comprising receptive free nerve endings were present 72.2% in the outer layer, 33.3% in the inner layer, and 65.0% in the subcutaneous tissue and SP-IR, 94.9% in the outer layer and 69.4% in the subcutaneous tissue of the rat TLF. The innervation observed consisted of fibers of passage and nerve endings, some of them go with blood vessels. The samples of rat TLF showed a dense innervation with a broad net of nerve fibers, and by comparing it with the human one, the authors concluded that the innervation density and the features of the free nerve endings was alike [20]. Barry et al. [9] observed that the muscles latissimus dorsi, gastrocnemius and erector spinae have the similar proportions of nerve fibers, but the TLF has three times higher density than the latissimus dorsi muscle (3.4 ± 0.6 and 1.0 ± 0.1 fibers per 40,000 µm^2^, respectively). Besides, the majority of the nerve fibers were found at the muscle-fascia interface. Likewise, the diameter of the nerves was established in Benetazzo et al. [14] with a mean of 15 µm (in human samples) and Corey et al. [27], ≤2 µm (in rats). The latter author also identified CGRP+ within the samples observed, being an 88% at T13 level and 73% at L1. Fede et al. [10] studied the TLF and gluteal fascia, demonstrating that both have a neural plexus inside, formed by free nerve endings both sensitive and autonomic, being positive to S100 and TH antibodies. Really, the TLF has a higher density of free nerve endings (9.01 ± 0.98%) respect to the gluteal fascia (2.78 ± 0.6%), whilst the density was similar in the different portions of TLF. The mean nerve length was 87.1 ± 1.0 mm in the TLF and 3.2 ± 0.6 mm in the gluteal fascia, while the thickness was respectively 5.8 ± 0.2 µm and 4.9 ± 0.2 μm. Besides, the authors also observed Golgi tendon organs close to the myofascial junctions, and muscle spindles in the perimysium connected with the fascia. Also, Marpalli et al. [18] studied the TLF and noted that the number of free nerve endings in the right sacral level, 12.762 ± 0.42, is bigger than in the right thoracic vertebral level (10.210 ± 0.374). Many authors [8,10,18,22,23,24,25] agree that the free nerve endings have the features of nociceptors. Nobody has found Pacini, Ruffini or Golgi-Mazzini corpuscles in the TLF samples. Mense and Hoheisel [23,24,25] in their studies described an increase of nociceptors (SP-IR), both in the density and in the length, in inflamed TLF, passing from a density of 2.0 per 1000 µm^2^ in health to 2.5 in inflamed one [23,24]. They observed the same effect in rats and humans. Two studies identified autonomic nerve fibers in the connective tissues of the low back and TLF (with a density of 0.08% TH positive fibers) [10,27]. Corey et al. [27] hypothesize that possibly they are Aδ and/or C fibers. There was only one study with respect to the trapezius fascia, in which they concluded there is a profuse innervation perforating the fascia with fibers from the peripheral and vegetative nervous systems [15]. Also, there was one study about the pectoralis major expansion fascia in which the mean number per area of free nerve endings, Pacini and Ruffini corpuscles were 25.59, 0.47 and 0.12, respectively. Some of those free nerve endings were perivascular [13].

There were four papers considering the upper limb. Stecco et al. [13] found various types of receptors with different densities as shown in the Table 2. Some of them were perivascular, as in the lacertus fibrosus. In another study, Stecco et al. [20] evaluated the innervation of the palmar aponeurosis and the flexor retinaculum, demonstrating that both are rich in free nerve endings. Also, there was a higher density of Golgi-Mazzoni and Pacini corpuscles in the palmar aponeurosis (median-Mdn = 2.0 in the palmar aponeurosis, and 0.0 in the flexor retinaculum). In addition, the authors compared the healthy palmar aponeurosis with those sampled by patients with Dupuytren’s disease, highlighting that free nerve endings are denser in distribution in the pathological fasciae (Mdn = 38.0 with respect to normal Mdn = 22.0). Two studies were performed on horse anterior limbs, one in the antebrachial fascia, the carpal flexor and extensor retinacula, and the metacarpal flexor retinaculum, describing a large number of nerve fibers in the areolar connective tissue, several of them surrounded by blood vessels, whilst in the dense connective tissue the presence of nerves was smaller. They did not find Pacini, Ruffini or Golgi-Mazzini corpuscles [12] in the horse retinacula, however in the ergot they found perivascular nerve bundles and Ruffini endings in its connective tissue and near the origin and insertions of the ligaments [11].

Regarding the lower limb, there were ten studies focused on fascial innervation. Fede et al. [21] studied the different soft tissues of the hip. The authors concluded that the superficial fascia was the second most densely innervated tissue (33.0 ± 2.5/cm^2^), the first being the skin (64 ± 5.2/cm^2^). The deep fascia presented a nerve density of 19 ± 5.0/cm^2^ and the nerves have a mean diameter of 15.5 ± 4.7 µm, the distribution of these nerve fibers differs from that of other types of tissues, forming sprouting networks along the fascial tissue. Taguchi et al. [28] described dense innervation in the distal third of the rats crural fascia. This innervation was significantly higher (number of fibers CGRP-IR 91.5 ± 11.3 and Peripherin-IR 65.0 ± 7.4) and longer (CGRP-IR 3171.4 ± 483.5 µm) than in proximal (CGRP-IR 51.0 ± 6.0, Peripherin-IR 36.7 ± 5.8 and CGRP-IR 1749.1 ± 189.8 µm) and middle (CGRP-IR 56.2 ± 10.4, Peripherin-IR 38.8 ± 3.3 and CGRP-IR 2041.3 ± 150.3 µm) parts. In addition, the nerve fibers were numerous in the lateral and medial parts and scarce in the center; 43% of the C-fibers observed were polymodal receptors, that are nociceptors responding to mechanical, chemical or heat stimuli, on the contrary, none of the Aδ-fibers were polymodal. Benetazzo et al. [14] observe nerve fibers concentrated in the middle and deep layers of the human crural fascia (estimated density 1.2% of all the area), and these fibers had a mean diameter of 12.1 ± 6.1 µm. Satoh et al. [17], found fascial nerves related to the packed collagen fibers within the popliteal fascia. Sanchis-Alfonso and Rosello-Sastre [19] described rich perivascular innervation on the knee lateral retinaculum, few nociceptive. They, additionally, observed big and small nerves, copious in neuromatous structures, with a pattern of neural sprouting within the fascia of painful knees. In the plantar fascia [16] were located several free nerve endings and Pacini and Ruffini corpuscles. They were prominent, in particular, in the medial, lateral and distal parts of the plantar fascia, where it is thinner and it merges with the abductor hallucis and digiti minimi fasciae, or where it is linked to the metatarsophalangeal joints. There was more innervation where a large number of fibers from the sole of the foot muscles are inserted in the inner surface, than in the outer surface of the plantar fascia. Three studies focused on the periostea, one found mechanosensitive nociceptors in the femur periosteum, specifically in its fibrous layer, where they could identify CGRP+ sensory fibers with densities of: 2664 ± 127 intersections/mm^2^, NF200+ sensory fibers: 2679 ± 233 intersections/mm^2^ and TH+ sympathetic fibers: 3138 ± 157 intersections/mm^2^ [7]. Other observed sensory neurons with the function of proprioceptors in the tibial periosteum [29]. Finally Thai et al. [9] located myelinated spinal afferent nerve endings, peptidergic and non-peptidergic unmyelinated (nociceptors) spinal nerve endings on both the femur and tibial periostea, obtaining the data summarized in Table 3. Tagged axons had multiple divisions forming a “net-like meshwork” upon an extensive area of the periosteum, some of the nerve endings travelled longitudinally and in parallel along the long axis of the bone.

## 4. Discussion

The purpose of this review was to collect, systematically present and analyze the existing evidence on muscular/deep fascia innervation from a histological and immunohistochemical point of view from 2000 to 2021. Twenty-three studies about this topic were found and reviewed. 

All the studies reported positive results in support of the presence of free nerve endings in all types of fasciae that were observed. Figure 2 depicts the areas and kind of innervation found. Two studies confirmed the presence of innervation forming a network permeating the fasciae [10,27]. There were many methodologic differences among studies. The most important variations were in sample, types of immunohistochemical markers, measurement methods, and fascial parts. However, the results of this review support the idea that the fasciae are very well innerved. 

The TLF (Figure 3) was the focus of most of the reports (8 out of 23) [8,10,14,18,22,23,24,25]. Low back pain occurs very frequently, so it is not surprising that the TLF is the most studied area in order to find out what its role is in this type of pain. And, given that, in all observations, a multitude of free nerve endings have been found forming a dense, mainly nociceptive, nerve network. Consequently, it is likely that TLF plays an important role in the generation and perception of low back pain [18,30,31]. Also, the fascia of the inferior limbs has been the subject of several studies [9,14,16,17,19,21,28,29], however, these reports have focused on different areas of the limb thus these data are not comparable. For the superior limbs, only 5 papers are present, and among them, two are in horses [16,17,18,20,26]. In reviewing the literature, no data was found on immunohistochemical studies about the innervation of the fasciae of the head, apart from masseter, of the anterior part of the trunk or neck areas, leading to the proposition that research should be carried out in these areas.

Three studies have compared the density of innervation of the deep fascia and of the underlying muscles, demonstrating in all the cases that the fasciae result significantly more densely innervated: TLF respect to latissimus dorsi [8], masseter fascia and muscle [26], quadriceps muscle and fascia lata [21]. This is in line with the suggestion that the fascia can be a sensory organ and that on many occasions myofascial pain could be caused by a fascial alteration, rather than a muscular problem. The possible causes of myofascial pain have been discussed in several articles: in some experimental studies it was shown that the hypertonic saline solution can cause a more intense and longer lasting pain when injected in fascia than in muscle [32,33]. Langevin et al. [34,35,36], have hypothesized that reiterated postures, sports, or repetitive motions may generate altered movement patterns, which increase the thickness of the tissue and in turn limiting the sliding between fascia layers. This situation would augment inflammation and pain increasing the stimulation of free nerve endings present in them. Other authors [37,38,39], postulate that new lines of force that may be generated after fascial adhesions due to trauma, overuse, or surgery, would modify the density of hyaluronan, increasing its viscoelasticity, which would produce changes in the activation of nerve receptors embedded in the fascia. If this activation exceeds their capacity of adaptation, they may become hyperactivated and generate pain.

It is interesting to note that in all the studies were observed mostly free nerve endings, often polymodal in nature, so their role can be either proprioception or nociception. In some cases, proprioceptors are specifically identified in tibial periosteum [29] and in TLF [8,10], whilst nociceptors were identified either in the TLF [8,18,21,22,25], low back fascia [27], masseter fascia [26], crural fascia [28], knee retinaculum [19] or femur and tibia periostea [7,9]. More debated is the presence of corpuscular receptors (Pacini, Ruffini, Golgi-Mazzini) in the deep fascia. Various immunohistochemical examinations document their absence in the anterior limb of the horse [12] and in the TLF [10,23,24,25]. In any case, the absence of corpuscles does not imply that the TLF does not have a proprioceptive role, as polymodal receptors can assume this role. On the other hand, these corpuscles are observed in the superior limb fasciae [13,20], in the flexor retinaculum of the wrist, palmar and plantar aponeurosis [13,16,20] in the ergot insertion of the horse [11]. This distribution may suggest that the encapsulated receptors are present in areas closer to the joints or in the extremities, where greater proprioceptive refinement is required, emphasizing the static/dynamic proprioceptive and/or motor coordination role of the fasciae [10,16,20]. It is suggested that further immunohistochemical studies regarding this would be made in this direction.

Another important finding is the particular distribution and precise localization of the innervation in the fasciae. While there is agreement that the innervation observed forms networks of nerve fibers distributed in all directions [21,22], the density and type of innervation is not homogeneous. Indeed, the sensory nerve fibers observed in femur and tibia periostea were situated in such a way passing longitudinally and in parallel to the axis of the bone, so that they can perceive mechanical distortion or stress along a particular axis of the mineralized bone or the periostea [7,9]. Similarly, in the crural fascia the nerve fibers are distributed so that mechanical nociception (Aδ fibers) occurs along the lateral border of the crural fascia, while polymodal nociceptive receptors (C fibers) are located only in the distal third [28]. On the other hand, the nerve distribution in the case of the wrist flexor retinaculum suggests that it is involved in signaling changes in joint volume as well as motor perception. Finally, in the case of the gluteal, pectoralis fasciae and lacertus fibrosus, the lower nerve density suggest a minor involvement of these fasciae in perception, but maybe they can play a role in the coordination of tension and mechanical transmission [10,13]. 

The most interesting findings were the congruence among all the studiesabout the increased nociceptor density or length in pathological fasciae, both at the level of the palmar aponeurosis in Dupuytren’s disease [20], in induced TLF inflammation [23,24,25], and in the knee painful retinaculum [19]. Besides, several authors [8,10,13,17,22,28], emphasize the importance of the fasciae as a nociceptive sensor, both in physiological and pathological conditions. Joint capsules and ligaments that had undergone a previous injury changing the proprioceptive representation of the fascial system, can contribute to ongoing instability, leading to repeated injuries, instability, inflammation, and pain [40]. The combination of inflammation and reduced mobility would lead to fibrosis, adhesions, and contractures, which, in turn, would increase tissue stiffness and difficulty of movement, and therefore, pain [35]. Additionally, it has been observed that the fasciae share innervation with other structures [15,23,25,29], which could cause a convergence of sensory inputs and an amplification of pain. As it been observed in Schilder et al. [41], the response to an electrical stimulus on the TLF is different from that on the muscle, with the former producing intense and long-term potentiation, leading the authors to consider that it could be related to the amplification of pain and the possible development of chronic pain. This may be because fascial nociceptors seem to be predisposed to sensitization to chemical and mechanical stimuli, this predisposition could also explain the acute pain [37,42]. These observations may support the hypothesis that this tissue may have part as a pain generator, and that the fasciae may experience peripheral sensitization, which adds to the central sensitization in explaining chronic pain. Despite these promising results, questions remain, such as the mechanism by which nerve fibers length increase in the inflamed TLF or how free nerve endings in other unstudied fasciae parts respond to the pathology. Further research should be undertaken to investigate these issues. 

These findings may be somewhat limited by the type of studied subjects. Indeed, the human samples are mostly taken from old subjects (60–96 years), except for 4 studies that were collected in young/adult subjects (15–50 years). and, therefore, the innervation observed may be subject to changes due to ageing, so it would be advisable to carry out more studies on samples of young subjects and to compare the characteristics of the innervation with those of older subjects. Secondly, the immunohistochemical markers that have been used are different, so both the description and quantification of innervation may be influenced for this reason. It would be useful to compare the various types of markers and the types of receptors in the same fascia to make more precise observations. Lastly, in each study reviewed, the innervation has been characterized differently, from the density to the length of the nerve fibers (Table 4), thus preventing a real comparison between the published data. Therefore, it can be suggested that protocols could be implemented to establish the type of immunochemical markers and how to characterize the innervation. Moreover, better characterize the innervation found in terms of density, length, and thickness could provide enough information to picture how is innervated the studied tissue as well as to compare data between studies.

## 5. Conclusions

According to the results of this review, the fasciae is a profusely innervated tissue; in particular, it contains proprioceptors and nociceptors, and the latter increase in density and/or length of pathological conditions. These findings contribute in several ways to our understanding of fascial innervation and provide a starting point for further research in this area. This new understanding should help to improve the management and therapeutic approach of pain.

## Figures and Tables

**Figure 1 ijms-23-05674-f001:**
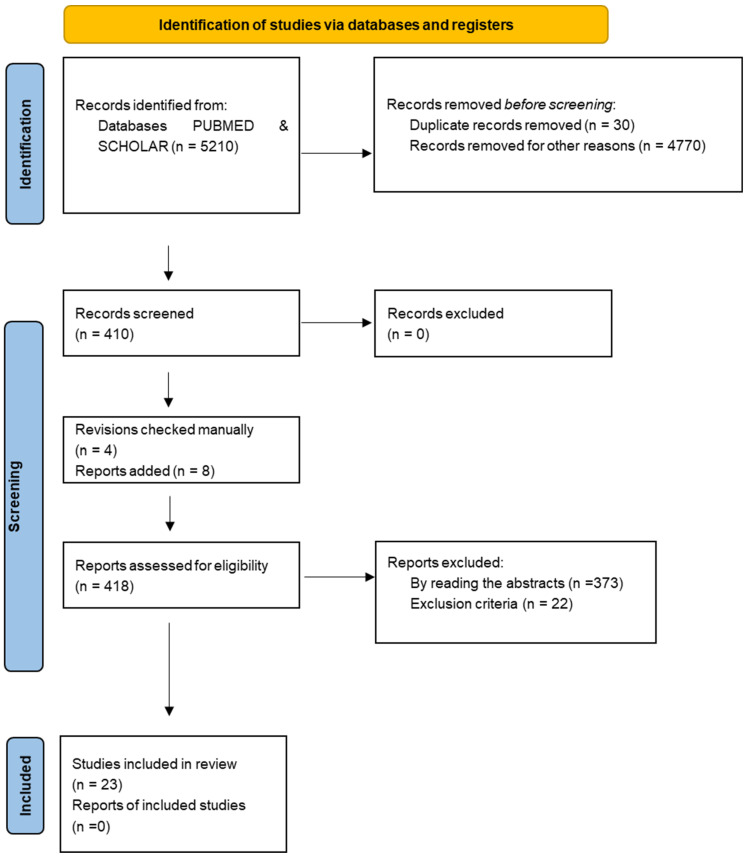
PRISMA flow diagram of article selection. From: Page MJ, McKenzie JE, Bossuyt PM, Boutron I, Hoffmann TC, Mulrow CD, et al. The PRISMA 2020 statement: an updated guideline for reporting systematic reviews [6].

**Figure 2 ijms-23-05674-f002:**
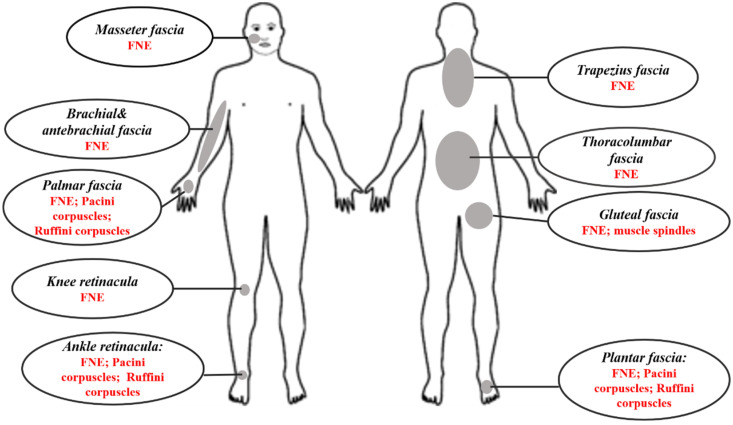
Graphical representation of the described sites and types of innervation.

**Figure 3 ijms-23-05674-f003:**
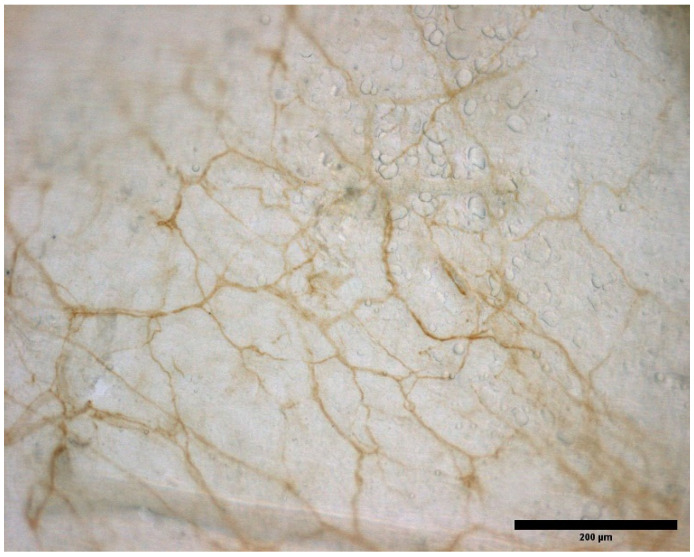
Innervation of a mouse TLF. Sample stained with S100.

**Table 1 ijms-23-05674-t001:** List of immunohistochemical markers, targets, and studies.

Marker	Detection	Study
SP	Nociceptors	Sanchis-Alfonso and Rosello-Sastre, 2000 [19]; Tesarz et al., 2011 [22]; Barry et al., 2015 [8]; Hoheisel, Rosner and Mense, 2015 [24]; Mense and Hoheisel 2016 [25]; Mense, 2019 [23]; Alhilou et al., 2020 [26]
PGP 9.5	Neuronal	Sanchis-Alfonso and Rosello-Sastre, 2000 [19]; Corey et al., 2011 [27]; Tesarz et al., 2011 [22]; Taguchi et al., 2013 [28]; Barry et al., 2015 [8]; Hoheisel, Rosner and Mense, 2015 [24]; Mense and Hoheisel 2016 [25]; Skalec and Egerbacher, 2017 [12]; Mense, 2019 [23]; Alhilou et al., 2020 [26]; Thai et al., 2020 [9]; Fede et al., 2021 [10]
CGRP	Nociceptors	Sanchis-Alfonso and Rosello-Sastre, 2000 [19]; Martin et al., 2007 [7]; Corey et al., 2011 [27]; Tesarz et al., 2011 [22]; Taguchi et al., 2013 [28]; Barry et al., 2015 [8]; Hoheisel, Rosner and Mense, 2015 [24]; Mense and Hoheisel 2016 [25]; Mense, 2019 [23]; Thai et al., 2020 [9]
NGF	Sensory and sympathetic neurons	Sanchis-Alfonso and Rosello-Sastre, 2000 [19]; Mense, 2019 [23]; Alhilou et al., 2020 [26]
TH	Postganglionic sympathetic fibers- dopaminergic and noradrenergic neurons	Martin et al., 2007 [7]; Tesarz et al., 2011 [22]; Hoheisel, Rosner and Mense, 2015 [24]; Mense, 2019 [23]; Thai et al., 2020 [9]; Fede et al., 2021 [10]
S100	Schwann cells	Sanchis-Alfonso and Rosello-Sastre, 2000 [19]; Stecco et al., 2007 [13]; Benetazzo et al., 2011 [14]; Domingo et al., 2011 [15]; Stecco et al., 2013 [16]; Skalec and Egerbacher, 2017 [12]; Lusi and Davies, 2017 [11]; Stecco et al., 2018 [20]; Fede et al., 2020 [21]; Fede et al., 2021 [10]
Neurofilament protein	Complex networks of axons	Sanchis-Alfonso and Rosello-Sastre, 2000 [19]
DiI	Tracer for neuronal and other cells	Gajda et al., 2004 [29]
NF200	Neuronal marker	Martin et al., 2007 [7]; Thai et al., 2020 [9]
MBP	Myelinating glia	Domingo et al., 2011 [15]
Peripherin	Peripheral neurons, including enteric ganglion cells	Taguchi et al., 2013 [28]
NSE	Neuron specific enolase	Barry et al., 2015 [8]
TRPV1	Epidermal and dermal cells, as well as free nerve fibers and Merkel cells	Mense and Hoheisel 2016 [25]; Mense, 2019 [23]
RT97	Neuronal marker	Satoh et al., 2016 [17]
Tubulin	Neurons	Stecco et al., 2018 [20]
NR2B	Glutamatergic neuron	Alhilou et al., 2020 [26]

SP: Substance P, PGP 9.5: Protein Gene Product, CGRP: Calcitonin Gene-Related Peptide, NGF: Nerve Growth Factor, TH: Tyrosine Hydroxylase, NF200: Neurofilament, MBP: Myelin Basic Protein, NSE: Neuron-Specific Enolase, TRPV1: Transient Receptor Potential Cation Channel Subfamily V Member 1, RT97: Neurofilament Monoclonal Antibody.

**Table 2 ijms-23-05674-t002:** Summary of the mean number of receptors per cm^2^ observed in the different areas of the superior limb.

Type of Receptors (Mean Number/cm^2^)	Brachial Fascia	Lacertus Fibrosus	Antebrachial Fascia	Flexor Retinaculum
Free nerve endings	48.57	27.36	44.37	53.55
Pacini corpuscles	0.43	0.26	0.26	0.66
Ruffini corpuscles	0.29	0.1	0.26	0.55

**Table 3 ijms-23-05674-t003:** Number of nerve fibers in periosteum in relation to immunohistochemical marker types according to Thai et al. (2020) [9].

Markers	CGRP+	CGRP-	NF200+	NF200-	NF200+ CGRP-	NF200+ CGRP+	NF200- CGRP+	NF200- CGRP-
N° of fibers	>20	5–20	5–20	>20	5–20	5	>20	>20

(+) Positive to the immunohistochemical marker. (-) Negative to the immunohistochemical marker.

**Table 4 ijms-23-05674-t004:** Nerve fibers measurements.

**Descriptive**	Sanchis-Alfonso and Rosello-Sastre, 2000 [19]; Gajda et al., 2004 [29]; Domingo et al., 2011 [15]; Stecco et al., 2013 [16]; Satoh et al., 2016 [17]; Skalec and Egerbacher, 2017 [12]; Lusi and Davies, 2017 [11]
**Number**	Stecco et al., 2007 [13]; Martin et al., 2007 [7]; Benetazzo et al., 2011 [14]; Taguchi et al., 2013 [28]; Thai et al., 2020 [9]
**Thickness**	Fede et al., 2021 [10]
**Length**	Taguchi et al., 2013 [28]; Hoheisel, Rosner and Mense, 2015 [24]; Mense and Hoheisel 2016 [25]; Mense, 2019 [23]; Fede et al., 2021 [10]
**Diameter**	Corey et al., 2011 [27]; Benetazzo et al., 2011 [14]; Fede et al., 2020 [21]
**Proportion**	Corey et al., 2011 [27]; Barry et al., 2015 [8]
**Density**	Corey et al., 2011 [27]; Tesarz et al., 2011 [22]; Barry et al., 2015 [8]; Stecco et al., 2018 [20]; Alhilou et al., 2020 [26]; Fede et al., 2020 [21]; Fede et al., 2021 [10]

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
