# Peer review of "Fascial Innervation: A Systematic Review of the Literature"

_ijms, 2022, doi:10.3390/ijms23105674_

Round 1

Reviewer 1 Report

This is a good manuscript, with the majority of reviewer comments are related to grammar corrections.

Line 14-replace “These days” with Today or Currently

Line20 it is mentioned 2000 to 2020 however Line 79 states 2000-2021. Please correct.

Lines 22,55,99 the word “made” should be replaced with “performed”

Line 26 Various areas have been studied with the thoralumbar fascia

Line 39 Replace “Really” with This effective

Line 57 Replace last years with In recent years

Line 65- “of the body have been studied to date. In addition, we included studies related…

Line 66 remove “about”

Line 69 remove “a”

Line 73 should be “GOOGLE SCHOLAR”  Limitation to use of only 2 databases is a weakness of this review, typically at least 3 are included, and yet use of 2 is acceptable particularly on this topic.

Line 89 Remove sentence “Information from the selected articles” (this is unnecessary information and addition of 2022 only adds confusion.

Line 121 “In the head, there is”

Note there is frequent use of “Authors”, the A should not be capitalized, but more importantly, sometimes it is unclear which authors (or studies) you are referring to. I strongly suggest reducing the use of the term Authors and use terms like “these studies”  “the latter study” or “Findings support” with appropriate citations to get away from the use of authors.

Line 142 established

Line 170 “as”

Capitalize the word (Table X) and (Figure X)

Line 241 replace probably with “likely”

Line 339 Moreover, better characterization of the innervation….

Reviewer 2 Report

In this paper titled “Fascial innervation: a systematic review of the literature”, Suarez-Rodriguez et al. conducted a systematic review on muscular/deep fascia innervation considering only the morphological and immunohistochemical papers from the literature search.

I found the work very interesting and well conducted. The topic is clearly stated and the review was carried on with a correct systematic approach. The search strategy is well constructed and the literature was qualitatively appraised and selected.

The evidences from the included literature were well presented and analysed. The findings of this review were extensively evaluated and discussed, and they can contribute to the understanding of fascial innervation and improve the pain relief strategies for myofascial syndromes.

I have only a few minor suggestions for the authors:

  1. Please check the entire text for typos or minor mistakes (e. double square bracket at line 52)
  2. Consider that one or more illustrations or figures could better support the arguments of the authors

Besides these minor points, I believe that the authors did a very nice job and the paper definitely deserves to be published in IJMS.
